# Design, Evaluation and Comparison of Nanostructured Lipid Carriers and Chitosan Nanoparticles as Carriers of Poorly Soluble Drugs to Develop Oral Liquid Formulations Suitable for Pediatric Use

**DOI:** 10.3390/pharmaceutics15041305

**Published:** 2023-04-21

**Authors:** Giulia Nerli, Lídia M. D. Gonçalves, Marzia Cirri, António J. Almeida, Francesca Maestrelli, Natascia Mennini, Paola A. Mura

**Affiliations:** 1Department of Chemistry, University of Florence, Via Schiff 6, Sesto Fiorentino, 50019 Florence, Italy; giulia.nerli@unifi.it (G.N.); marzia.cirri@unifi.it (M.C.); francesca.maestrelli@unifi.it (F.M.); 2Research Institute for Medicines (iMed.ULisboa), Faculty of Pharmacy, Universidade de Lisboa, Av. Prof. Gama Pinto, 1649-003 Lisboa, Portugal; lgoncalves@ff.ulisboa.pt (L.M.D.G.); aalmeida@ff.ulisboa.pt (A.J.A.); natascia.mennini@unifi.it (N.M.)

**Keywords:** chitosan nanoparticles, lipid nanoparticles, cefixime, oral liquid formulations for children, gastric stability, storage stability, cytotoxicity

## Abstract

There is a serious need of pediatric drug formulations, whose lack causes the frequent use of extemporaneous preparations obtained from adult dosage forms, with consequent safety and quality risks. Oral solutions are the best choice for pediatric patients, due to administration ease and dosage-adaptability, but their development is challenging, particularly for poorly soluble drugs. In this work, chitosan nanoparticles (CSNPs) and nanostructured lipid carriers (NLCs) were developed and evaluated as potential nanocarriers for preparing oral pediatric solutions of cefixime (poorly soluble model drug). The selected CSNPs and NLCs showed a size around 390 nm, Zeta-potential > 30 mV, and comparable entrapment efficiency (31–36%), but CSNPs had higher loading efficiency (5.2 vs. 1.4%). CSNPs maintained an almost unchanged size, homogeneity, and Zeta-potential during storage, while NLCs exhibited a marked progressive Zeta-potential decrease. Drug release from CSNPs formulations (differently from NLCs) was poorly affected by gastric pH variations, and gave rise to a more reproducible and controlled profile. This was related to their behavior in simulated gastric conditions, where CSNPs were stable, while NLCs suffered a rapid size increase, up to micrometric dimensions. Cytotoxicity studies confirmed CSNPs as the best nanocarrier, proving their complete biocompatibility, while NLCs formulations needed 1:1 dilution to obtain acceptable cell viability values.

## 1. Introduction

Development of drug formulations tailored for pediatric use is one of the main challenges for the pharmaceutical industry and regulatory agencies, due to the growing needs for accessible age-appropriate pediatric medicines able to ensure a safe and effective adherence to the prescribed treatment [1]. Only about 20% of prescribed drugs have dosage forms that are proper for pediatric patients [2,3]. Additionally, health care providers are frequently forced to the off-label use of adult dosage forms by manipulating them, e.g., by extemporarily crushing tablets, opening capsules or diluting solutions [4,5,6]. These actions could negatively affect the chemical-physical and microbiological stability, as well as lead to mistakes in terms of uncontrolled dose accuracy, resulting in risks of toxicity and unknown bioavailability of the obtained preparations [7,8].

The development of pediatric products is considerably challenging due to the specific requirements of this special patient population [9]. In particular, from a clinical pharmacology point of view, children not only have absorption-distribution-metabolism-elimination processes different from adults, but they are also a heterogeneous cluster comprising various sub-categories. The European Medicine Agency classifies the pediatric population into five age groups, each one with different requirements when considering both the type of formulations and administration route [10]. Furthermore, pediatric formulations must satisfy more features than those for adults. In particular, acceptability, i.e., the “ability and willingness of the patient to use, and its caregiver to administer, the medicine”, has been identified by 90% of pediatricians as a critical attribute to ensure patient adherence to the therapy [11,12].

The oral route is considered the most convenient drug administration way in the pediatric field due to its several advantages over other delivery pathways, including a wide variety of dosage forms, easy self-administration, good patient compliance, and a lack of pain sensation [10]. However, it must be kept in mind that the actual therapeutic efficacy of the drugs depends on their oral bioavailability, which mainly relies on their physical-chemical and biopharmaceutical properties, such as solubility, dissolution rate, permeability, and stability. Unfortunately, several active compounds have a very low aqueous solubility, leading to issues of erratic/low oral absorption and limited/variable oral bioavailability, then presenting a major challenge to the development of effective pharmaceutical formulations [13]. Several strategies have been investigated during the last few decades to overcome this very important issue [14,15]. Among these, nanoparticle (NP)-based delivery systems recently emerged as a promising approach for improving the gastrointestinal absorption of low-soluble drugs [16], including in the pediatric field [17]. In fact, entrapment into NPs can enhance solubility, bioavailability, permeability, and stability of pediatric drugs, thus making it possible to more efficiently achieve the therapeutic dose and reduce side effects [18]. Moreover, NP formulations can be tuned to obtain a prolonged release, achieving a drug release over time and reducing the number of administrations [18]. Among the several kinds of NPs, those based on natural polymers or lipid materials are the most promising candidates, particularly for pediatric formulations, since their components are biocompatible, biodegradable, and non-toxic, and they are good candidates to entrap poorly water-soluble drugs. It must also be considered that liquid formulations are deemed as the most suitable oral dosage form for pediatric patients, due to their higher administration ease and simplicity of dosage/weight adaptability [10]. Among liquid formulations, solutions are given preference over suspensions, in virtue of the absence of choking risks or dosing errors due to a not correct resuspension of the sediment, and are also better in enteral feeding administration.

Based on these premises, our general purpose was the development of effective oral solutions of poorly soluble drugs suitable for pediatric patients, in order to limit the off-label use of adult dosage forms and of extemporaneous preparations, with all the related issues described above. With this aim, in the present work, we developed and compared two different kinds of NPs, i.e., nanostructured lipid carriers and chitosan nanoparticles as possible nanocarriers for the development of oral pediatric solutions of cefixime (CEF), a third-generation cephalosporin. It was selected as a poorly soluble model drug because of its wide use in pediatric treatments (being in the antibiotics group of the WHO List of Essential Medicines for children [19]), as well as for the lack of formulations specific for the pediatric population and of commercially available oral solutions. The entrapment of CEF into suitable lipid or polymeric NPs should allow the successful development of liquid formulations of the drug [18], overcoming the issue of its very low aqueous solubility that is also considered the key parameter responsible for its low oral bioavailability [20,21].

Nanostructured lipid carriers (NLCs) are nanoparticles consisting of a mixture of biocompatible liquid and solid lipids, dispersed in an aqueous phase. They constitute an evolution with respect to the original solid lipid nanoparticles (SLNs), since the presence of the liquid lipid leads to the formation of imperfections in the structure of the solid lipid matrix, resulting in a series of benefits, including better loading capacity, enhanced physical stability, and less premature drug leakage [22,23]. NLCs emerged as effective carriers to improve the oral bioavailability of poorly soluble hydrophobic drugs by enhancing their apparent solubility in the gastrointestinal tract, improving their stability and favoring their permeation across the intestinal membrane [24,25]. In addition, the large-scalable production methods and the use of safe and biocompatible components make NLCs an ideal candidate carrier for the development of drug delivery systems [23].

On the other hand, among NPs based on natural polymers, chitosan (CS) NPs have been widely and successfully used as oral carriers for several kinds of both hydrophilic and hydrophobic drugs [26]. CS is a linear polysaccharide produced by complete or partial deacetylation of chitin, and it consists of randomly distributed β-(1,4)-linked D-glucosamine and N-acetyl-D-glucosamine units. The wide applications of CS NPs are due to the numerous favorable properties of this polymer, such as biocompatibility, biodegradability, absence of toxicity, mucoadhesive ability, and permeation enhancing ability, joined to antimicrobial, anti-inflammatory, and wound-healing effects [27]. Moreover, CS NPs proved the benefits of sustaining and controlling the release of drugs, enhancing their solubility and stability [28]. Therefore, also in consideration of their straightforward production methods, particularly by ionotropic gelation, they were chosen as a possible CEF carrier as an alternative to NLCs.

Different NLCs and CS NPs formulations have been developed and characterized in terms of mean size, polydispersity index, and Z-potential values, and the effect of drug loading on such parameters was also evaluated. The best formulations have been further characterized and compared for drug entrapment and loading efficiency, release behavior, intra-gastric and storage stability, and assayed for cytotoxicity.

## 2. Materials and Methods

### 2.1. Materials

Precirol^®^ ATO5 (glyceryl distearate), Transcutol^®^ HP (highly purified diethylene glycol monoethyl ether), Transcutol^®^ P (purified diethylene glycol monoethyl ether) Compritol^®^ 888ATO (glyceryl dibehenate), Gelucire^®^ 48/16 (polyoxyl-32 stearate (type I), Geleol^®^ (glycerol monostearate), Labrafac^®^ Lipophile WL 1349 (caprylic triglyceride, medium chain triglycerides), Labrasol^®^ (caprylocaproyl polyoxyl-8 glycerides), Labrasol^®^ ALF (caprylocaproyl polyoxyl-8 glycerides), and cetyl palmitate were kindly provided by Gattefossé (Saint-Priest, Cedex, France). Imwitor^®^ 491 (glyceryl monostearate), Imwitor^®^ 988 (glyceryl monocaprylate), Miglyol^®^ 810N (caprylic triglyceride), and Miglyol^®^ 812 (caprylic triglyceride) were supplied by IOI Oleo GmbH (Hamburg, Germany). Glyceryl tripalmitate, Pluronic^®^ F68 (poloxamer 188), stearylamine, low-molecular weight chitosan (CS, 50–190 kDa, 75–85% deacetylation degree), tripolyphosphate, 3(4,5-dimethyl-2-thiazolyl)-2,5-diphenyl-2H-tetrazolium bromide (MTT), propidium iodide (PI), sodium dodecyl sulphate (SDS), and dimethylsulfoxide (DMSO) were purchased from Sigma-Aldrich (St. Louis, MO, USA). Cefixime trihydrate (CEF) was a kind gift from Menarini (Florence, Italy). Purified water was obtained by inverse osmosis (Elix 3, Millipore SAS, Molsheim, France). All other chemicals were of analytical grade.

### 2.2. Screening of Solid and Liquid Lipids for NLC Preparation

Lipids selection was based on their solubilizing power towards the drug. The CEF solubility in solid lipids was determined according to Patel et al. [22] by adding a fixed amount of the drug (10 mg) to different amounts (from 200 up to 500 mg) of lipid heated at 80 °C (about 10 °C above the lipid melting point). The CEF solubility in liquid lipids was determined by adding 10 mg of the drug to a range from 200 up to 500 µL of the liquid lipids followed by a vigorous stirring and heating at 80 °C. The solubility of CEF was assessed visually, controlling the absence of drug crystals and the presence of a clear and homogeneous solution [29]. Each determination was performed in triplicate.

### 2.3. NLCs Preparation

NLCs were prepared by hot high-shear homogenization, according to Gonçalves et al. [30] and Cirri et al. [31]. Briefly, the selected solid (Precirol^®^ ATO5, alone or in mixture with Gelucire^®^ 48/16 or stearylamine) and liquid (Transcutol^®^ HP) lipids were put in a test tube. The aqueous phase consisted of 10 mL of aqueous solution containing Pluronic^®^ F68 (0.1% *w*/*v*) selected as non-ionic hydrophilic surfactant [31]. Both aqueous and lipid phases were placed in a water bath (WTB6, VWR, Darmstadt, Germany) at 80 °C (10 °C above the solid lipid melting point). After the fusion of the solid lipid, the aqueous phase was added to the lipid one and the system was stirred at 12,500 rpm for 5 min by a high-shear laboratory mixer (Silverson SL5M, Silverson LDT Machines, Chesham, UK). The dispersion was then removed from the water bath and cooled in an ice bath at 4 °C under gentle agitation, to obtain the solidification of the lipid matrix and produce NLCs. In the case of drug-loaded NLCs, CEF (10, 20, or 40 mg) was always added to the lipid phase to obtain a drug concentration in the final nanodispersion of 0.1, 0.2, or 0.4 mg/mL. Each batch has been prepared in triplicate. All the formulations were stored for no more than a week at 4 °C for further investigations.

### 2.4. Chitosan NPs Preparation

Chitosan (CS) NPs were prepared by ionic gelation technique using a previously optimized method [32], which was slightly modified. Briefly, CS and tripolyphosphate (TPP) stock solutions at 10 mg/mL in 1% *v*/*v* acetic acid and in ultra-pure water, respectively, were individually diluted to the desired concentrations (CS solution was diluted using a Pluronic^®^ F68 0.1% *w*/*v* aqueous solution). The TPP solution was added to the CS one, and the obtained mixture was homogenized under mild stirring, by up-and-down pipetting, at room temperature. NPs immediately formed after TPP addition to the CS solution at the fixed 1:5 *v*/*v* ratio. For preparing loaded NPs, CEF (10, 20, or 40 mg) was added to the CS solution before TPP addition to obtain a drug concentration in the final nanodispersion of 0.1, 0.2, or 0.4 mg/mL, respectively. Each batch has been prepared in triplicate. All the formulations were stored for no more than a week at 4 °C for further investigations.

### 2.5. Particle Size, Polydispersity Index, and Zeta Potential

The determination of mean particle size and polydispersity index (PDI) of NLCs and CS NPs formulations was made by Dynamic Light Scattering (DLS) (Zetasizer Nano S, Malvern Instruments, Malvern, UK). The samples were previously diluted with ultra-pure water to avoid multi-scattering phenomena. Particle size was expressed as a hydrodynamic diameter (nm). All measurements were made in triplicate at 25 ± 0.1 °C and the average values were calculated. The determination of the surface electrical charge (expressed as Zeta Potential) was performed by measuring the electrophoretic mobility of the particles (Zetasizer Nano Z, Malvern Instruments, Malvern, UK). Samples were diluted with ultra-pure water. All measurements were made in triplicate at room temperature and the average values were calculated.

### 2.6. Drug Quantification

The CEF assay was performed by UV analysis at λ_max_ of 290 nm using a microplate spectrometer reader (FLUOstar Omega, BMG Labtech GmbH, Ortenberg, Germany). It was controlled so that the used lipids and other components did not interfere with the UV spectrophotometric assay of the drug. A calibration curve (r^2^ = 0.999) was plotted over a CEF concentration range of 100–0.39 µg/mL. The limit of detection and limit of quantification were 0.12 and 0.39 µg/mL, respectively.

### 2.7. Entrapment Efficiency and Loading Efficiency

In the case of NLCs, entrapment efficiency (EE%) and loading efficiency (LE%) were determined by a direct method. Firstly, 2.5 mL of NLCs dispersion was purified by size-exclusion chromatography on Sephadex G-25/PD-10 columns (GE Healthcare, Solingen, Germany) in order to remove the non-encapsulated drug. Then, 500 µL of purified NLCs dispersion was treated with 500 µL of acetonitrile:water 1:1 *v*/*v* solution to disrupt the nanoparticles. After 5 min of centrifugation at 13,400 rpm (MiniSpin, Eppendorf, Hamburg, Germany), the drug content in the supernatant was quantified by UV assay at 290 nm, as described above.

In the case of CS NPs, EE% and LE% were determined by an indirect method. The concentration of the free drug present in the aqueous phase was determined after separation from the loaded NPs. Briefly, 1 mL of each sample was centrifuged (MiniSpin, Eppendorf, Hamburg, Germany) at 13,400 rpm for 5 min; afterwards, the supernatant, containing the non-encapsulated drug, was collected and quantified by UV assay at 290 nm as described above.

EE% of CEF was then calculated according to the following equation:(1)$$\mathrm{EE}\%=\frac{W_{entrapped\;drug}}{W_{total\;drug}}\times 100$$
where W*_total drug_* is the amount of drug initially used and W*_entrapped drug_* is the amount encapsulated.

Loading efficiency (LE%) was instead calculated according to the following equation:(2)$$LE\%=\frac{W_{entrapped\;drug}}{W_{excipients}}\times 100$$
where W*_entrapped drug_* is the amount of drug encapsulated and W*_excipients_* is the weight of the excipients used for NLCs or CS NPs preparation.

Each experiment has been performed in triplicate and the average values were calculated.

### 2.8. In Vitro Drug Release

In vitro drug release experiments from lipid and polymeric nanoparticles have been carried out according to the dialysis membrane method [33]. The dialysis bags (Spectra/Por^®^ Biotech-Grade Cellulose Ester dialysis membranes, MWCut-Off: 100 kDa, Repligen, Waltham, CA, USA) were filled with 1 mL of sample and immersed under stirring (300 rpm) for 2 h in 15 mL of simulated gastric fluid without pepsin (SGF) at 37 °C, composed from 2.0 g/L NaCl adjusted with HCl 0.1 N at pH 1.2 or at pH 4.5 (this last miming of the gastric pH of infant age < 12 months) [34], and for 4 h in 15 mL of simulated intestinal fluid (SIF) at pH 6.8 (6.80 g/L KH_2_PO_4_ and NaOH 0.89 g/L) at 37 °C. Samples of 200 µL were taken at predetermined intervals up to 360 min from the receiver solution and replaced by fresh medium, in order to guarantee sink conditions. The amount of released drug was determined by UV assay at 290 nm. A correction for the cumulative dilution was applied. Each experiment was performed in triplicate and the average values were calculated. The colloidal dispersions were purified before the experiments to remove the free drug and thus to be able to evaluate the actual effect of the different formulations on the drug release profile. NLC samples were purified by size-exclusion chromatography, while CS NP samples were centrifuged. The precipitated NPs were separated from the supernatant and finally resuspended in aqueous solution containing 0.1% (*w*/*v*) Pluronic^®^ F68.

### 2.9. Stability Studies under Simulated Gastric Conditions

The intra-gastric stability of loaded NLCs and CS NPs was evaluated. Simulated gastric fluid (SGF) without pepsin at pH 1.2 and 4.5 was prepared as previously described (see Section 2.8). The ratio of SGF solution:NLCs (or CS NPs) dispersion was 1:1 *v*:*v* [35]. The mixture was kept at 37 °C for 2 h. At defined time intervals, the particle size distribution and PDI of the samples were measured by DLS as described above (see Section 2.5). Each experiment was performed in triplicate and the average values were calculated.

### 2.10. Storage Stability Studies

The selected formulations of NLCs and CS NPs (both empty and drug-loaded) were stored for 3 months at 4 ± 1 °C and weekly checked for mean particle size, polydispersity index (PDI), and Zeta potential by DLS as described above (see Section 2.5). Samples were also subjected to visual inspection in order to verify eventual formation of mold or the appearance of aggregation or precipitation phenomena. At the end of the storage period, drug-loaded formulations were also examined for drug EE%.

### 2.11. In Vitro Cytotoxicity Studies

Caco-2, a human epithelial colorectal adenocarcinoma cell line (obtained from American Type Culture Collection, ATCC^®^ HTB-37™, Rockville, MD, USA), was used for cytotoxicity studies of the developed formulations. Cytotoxicity was evaluated by the MTT reduction and propidium iodide exclusion assays, according to a previously established procedure [36].

Cell viability was assessed after 24 h of incubation of the Caco-2 cell line with the different samples. This incubation time was chosen based on previous studies [30,36,37]. Briefly, the day before the experiments, after thawing the cell bank, cells within 2–5 passages number were seeded in sterile flat bottom 96-well tissue culture plates (Greiner Bio-One, Kremsmünster, Austria), at a cell density of 2 × 10^5^ cells/mL, 100 μL per well, in RPMI 1640 culture medium added with 10% fetal bovine serum, 100 units/mL of penicillin G sodium salt, 100 μg/mL of streptomycin sulphate, and 2 mM L-glutamine (all reagents were from Thermo Fisher, Waltham, MA, USA). After 24 h incubation at 37 °C and 5% CO_2_, the culture medium was substituted by fresh medium containing the various samples to be tested, each of which was analyzed in four wells per plate in three independent plates. The culture medium was used as the negative control, while sodium dodecyl sulphate (SDS, 1 mg/mL) was used as the positive control. After 24 h incubation, the medium was substituted with a 0.3 μM propidium iodide solution (stock solution 1.5 mM in DMSO, diluted 1:5 with culture medium). Fluorescence (excitation wavelength, 485 nm; emission wavelength 590 nm) was determined using a Microplate Reader (FLUOstar Omega, BMG Labtech GmbH, Ortenberg, Germany).

The MTT assay was then carried out by replacing the medium with a medium containing 0.25 mg/mL MTT. After 3 h incubation, the medium was removed and the intracellular crystals of formazan (a water-insoluble purple compound formed by MTT reduction by viable cells) were dissolved and extracted with DMSO. After 15 min, the absorbance at 570 nm was determined using a Microplate Reader (FLUOstar Omega, BMG Labtech GmbH, Ortenberg, Germany).

In the first case, data were expressed as propidium iodide uptake, according to the following equation:(3)$$propidium\;iodide\;uptake=\frac{{Fluorescence}_{sample}}{{Fluorescence}_{control}}$$
where *Fluorescence_sample_* is the fluorescence intensity (FI) of cells treated with NLCs or CS NPs samples and *Fluorescence_control_* is the FI of cells incubated with culture medium.

For the MTT assay, data were expressed as a % of viable cells with respect to cells exposed to the negative control, calculated according to the following equation:(4)$$cell\;viability\;\left(\%\;of\;control\right)=\frac{{Abs}_{sample}}{{Abs}_{control}}\times 100$$
where *Abs_sample_* is the absorbance at 570 nm of cells treated with NLCs or CS NP samples and *Abs_control_* is the absorbance of cells incubated with culture medium.

### 2.12. Statistical Analysis

Statistical analysis of the experimental data was carried out using the GraphPad Prism software version 6.0 (San Diego, CA, USA). All the data were expressed as the mean ± standard deviation (SD). One-way analysis of variance (ANOVA) with Student–Newman–Keuls comparison post hoc test was utilized to evaluate all statistically significant differences. A *p* value < 0.05 was considered significant.

## 3. Results and Discussion

### 3.1. Screening of Solid and Liquid Lipids for NLCs Preparation

NLCs were selected as potential suitable carriers for the development of CEF pediatric oral solutions, considering their biocompatibility and proved ability to improve solubility and bioavailability of poorly soluble drugs [24,25].

As a first step in the development of the CEF NLCs formulations, the solubilizing ability towards the drug of different solid and liquid lipids, selected considering their physiological tolerance and absence of toxicity [38], was evaluated. Drug solubility in lipids is an important factor in the selection of the better components for the development of an effective nanocarrier formulation, since it is directly correlated to the NLCs encapsulation efficiency and drug loading. A preliminary screening was carried out by adding a constant amount of drug (10 mg) to different amounts of melted solid or heated liquid lipid and visually assessing the formation of a transparent homogeneous system.

The results, expressed in terms of turbidity or transparency of the systems, are collected in Table 1. Among the examined liquid lipids, only Transcutol^®^ HP and P enabled the complete drug dissolution, reaching CEF concentrations up to 50 mg/mL. Transcutol^®^ (diethylene glycol monoehtyl ether) was then selected as liquid lipid for the preparation of NLCs. In particular, Transcutol^®^ HP (Highly Purified) was preferred to the P type due to its higher purity (99.9%). In fact, it has been demonstrated that the toxicity of Transcutol^®^ emerged in nonclinical studies carried out before 1990 was wrongly related to high levels of this solvent, while it should rather be attributed to the presence of significant amounts of impurities, especially ethylene glycol; as a confirmation of this, extensive literature data proved its safe use as a vehicle and solvent for multiple administration routes [39].

On the contrary, solubility studies were not useful to select the best solid lipid, since none of the examined solid lipids allowed a complete solubilization of CEF, even at the highest amount used (500 mg). Therefore, based on literature data, Precirol^®^ ATO5 (HLB 2) was selected since it is one of the most used solid lipids for NLCs preparation [18]. Moreover, the effect of the addition of Gelucire^®^ 48/16 was investigated, considering its high HLB (calculated/practical HLB 16/12 [40]) and its non-ionic surfactant nature, which should concur to improve the drug solubility and simultaneously promote the steric stabilization of the nanoparticulate system [41]. Finally, the effect of the addition of stearylamine was also evaluated, since, due to its positive charge, it could also contribute to improve the physical stability of the colloidal dispersion.

### 3.2. Preparation and Characterization of NLCs

A first series of NLCs formulations was initially prepared using different amounts and combinations of the selected solid lipids, while keeping constant the amount of both the selected liquid lipid Transcutol^®^ HP (5 µL/mL) and the drug (0.1 mg/mL), as reported in Table 2. The effect of increasing the drug loading concentration up to 0.2 and 0.4 mg/mL has also been tested on the formulation selected as the best in order to obtain final aqueous colloidal dispersions containing 10, 20, or 40 mg/100 mL of CEF.

The drug concentration to be loaded was determined considering its solubility in the formulation (see Section 3.1) and its minimum inhibitory concentration (MIC) necessary to exploit its antimicrobial activity. In a previous study [42], we found that the MIC of CEF against *Escherichia coli* is <0.5 µg/mL. Thus, even the lowest drug concentration used for NLCs loading (0.1 mg/mL) was more than 20 folds higher than its MIC value. Each batch was prepared both empty and drug-loaded, and adequately characterized, to investigate the influence of the drug encapsulation on the NLC properties (Table 3).

Empty NLC1 and NLC2 batches, containing only Precirol^®^ ATO5 as a solid lipid (at a concentration of 5 or 10 mg/mL, respectively), showed homogeneous size distribution of nanoparticles (PDI < 0.25), with a mean diameter around 154 or 203 nm, respectively, and a highly negative surface charge (Z-pot-31 or −25 mV, respectively), indicative of good colloidal stability. However, in both cases, CEF encapsulation negatively affected the systems, causing a loss of stability of the formulation, with aggregation phenomena. This was coupled with a sharp reduction of the negative surface charge of the nanoparticles (Z-pot around −10 mV) that could be attributed to the protonated amino group of CEF.

In the attempt of increasing the stability of the NLCs dispersion, and reduce the negative effect of CEF incorporation, Gelucire^®^ 48/16 was then added (20 mg/mL), maintaining the Precirol^®^ ATO5 concentration at its highest value (10 mg/mL). It was expected that this lipid component, due to its high HLB and non-ionic surfactant nature [40], could offer a steric stabilization effect, preventing nanoparticles aggregation in the colloidal system [41].

The addition of CEF in the obtained NLC3 formulation actually only gave rise to a moderate increase of the nanoparticle size, compatible with the drug incorporation, without causing aggregation phenomena. However, the very low Zeta potential (around −5 mV) of this formulation was judged not suitable to assure an adequate stability over time.

Considering the issues presented by these series of batches, stearylamine was added to the formulation with the aim of reversing from negative to positive the surface charge of NLCs and consequently avoid the observed reduction of the negative Zeta potential as a consequence of CEF addition. When comparing the data obtained with the empty and drug-loaded formulations (NLC4_E_ and NLC4_DL1_), we can see that the addition of the drug only caused the expected slight increase in nanoparticle dimensions, without adversely affecting PDI (whose value around 0.3 was considered indicative of a satisfying homogeneous population of vesicles [43]) or modifying the Zeta potential, which remained >+50 mV, indicating that this formulation presents the best lipid composition for obtaining NLCs with the desired physicochemical properties. Then, the effect of adding increasing amounts of drug was studied by keeping this last lipid composition constant. It was found that, even raising drug concentration up to 0.2 and 0.4 mg/mL (batches NLC4_DL2_ and NLC4_DL4_, respectively), the particle size, PDI, and Zeta potential of the colloidal dispersions remained substantially similar, indicating the good robustness of the developed NLCs formulation. The drug solubility in Transcutol^®^ HP (>50 mg/mL of lipid) was not exceeded even in these formulations, which appeared perfectly limpid when subjected to visual inspection.

The ability of the selected NLC4 formulation to entrap the active substance was then evaluated in terms of entrapment efficiency and drug loading (Table 3). The EE% of the selected NLC formulations ranged from about 36.5% for both NLC4_DL2_ and NLC4_DL4_ up to about 40% for NLC4_DL1_. Nevertheless, focusing instead on LE%, the batch NLC4_DL4_ clearly showed the best value, which was 3.7 or 2.1 folds higher than those of NLC4_DL1_ and NLC4_DL2_, respectively, due to the greater amount of drug initially added in this formulation. Therefore, considering the similar size, PDI, and Zeta potential values, but the highest LE%, the NLC4_DL4_ formulation was the one selected for further experiments.

### 3.3. Preparation and Characterization of CS NPs

CS NPs were selected as the alternative CEF nanocarrier to be compared with NLCs. CS was chosen as the natural polymer for NPs preparation, based on its biocompatibility, biodegradability, permeation enhancing properties, and almost total lack of toxicity [24] combined with its well-known ability to form nanoparticles [27,28,44]. CS NPs proved to also be suitable carriers for poorly soluble drugs [26,45,46,47,48]. Moreover, the use of CS NPs as carriers for the development of liquid oral CEF formulations for pediatric use appeared particularly appropriate, considering the CS antifungal and antimicrobial properties [49].

Among the several methods to produce CS NPs, ionic gelation is the best known and most-frequently used technique, due to its simplicity, low time consumption, mild processing conditions, absence of toxic reagents, and ease of production scale-up [50]. A previously optimized procedure was used [32], slightly modified with the use of a Pluronic F68 0.1% *w*/*v* solution to appropriately dilute the CS solution. This non-ionic surfactant was added as a stabilizer of the polymeric NPs [51] and it was used at the same concentration as in NLC formulations to allow a better comparison of the two kinds of nanoparticulate formulations. Even drug concentrations were the same used in the selected NLCs formulation. The composition of the different CS NP formulations tested is shown in Table 4, while their physicochemical properties are reported in Table 5.

In the case of CS NPs, the addition of CEF, even at the highest concentration tested (0.4 mg/mL), did not substantially change the properties of the nanocarriers in terms of size, PDI and, in particular, of Zeta potential, which remained always very high, assuring the good stability of the colloidal dispersions. However, PDI values of CS NPs were clearly greater than those of NLCs, indicative of a less homogeneous particle size distribution. This can be attributed to their different preparation method, since CS NPs production did not involve any high-speed homogenization step, but just a manual gentle blending of the system after the TPP addition. However, although there are no specific criteria established by regulatory agencies regarding the PDI values deemed acceptable, the obtained PDI values around 0.5 can still be considered admissible for oral delivery systems, while PDI values > 0.7 could not be accepted, since they would be indicative of a very broad particle size distribution [43].

Interestingly, in contrast to what observed for NLCs formulations, a significant increase not only in LE% but also in EE% was found by increasing CEF concentration from 0.1 to 0.2 up to 0.4 mg/mL. Finally, visual inspection of all the systems ensured the absence of drug crystals, even at the highest concentration used, confirming that the drug was completely dissolved in the CS solution.

Therefore, based on these results, CS NP_DL4_ formulation was selected for further experiments, due to its best EE% and LE% values.

### 3.4. In Vitro Drug Release Studies

The results of release studies of CEF in simulated gastric (SGF) and intestinal (SIF) fluids from the selected NLC4_DL4_ and CS NP_DL4_ formulations are shown in Figure 1. Since CEF is widely used in various stages of the children growth, the developed formulations have been tested at two different gastric pH values, i.e., 1.2 and 4.5, where the second one simulates the pH of infants of less than 12 months [34]. In fact, these different gastric pH conditions must be considered, because they could affect the release profile of the loaded drug from the system.

Regarding the NLC4_DL4_ formulation behavior, release studies performed using SGF pH 1.2 (Figure 1A) showed a high drug release (about 75%) in the first 2 h in the gastric medium, achieving the complete drug release within 6 h. Experiments performed using SGF pH 4.5 (Figure 1B) showed instead a slower drug release, reaching about 50% after 2 h in the gastric medium and about 70% after further 4 h in the intestinal medium. Such a different release behavior was related to the different pH of the gastric medium: the results seemed to indicate that the higher the acidity of the medium, the faster its penetration through the external shell of the NLCs, and, consequently, the drug diffusion. In contrast, the drug release behavior from CS NP_DL4_ was less affected by the different gastric pH conditions, and the release rate was much slower than from NLC4_DL4_ formulation, reaching only 22% and 30% after 2 h at pH 1.2 or 4.5, respectively, and only about 30% or 40%, respectively, after 6 h. Such results could be related to the particular CS properties. In fact, at low pH values (1.2–4.5), the high protonation degree of its amino groups (pKa 6.5) enhances the interactions with the cross-linker TPP. As a result, the drug was more strictly retained within the NPs compact structure, and then more slowly released. The fact that the release also remained slow in the simulated intestinal medium could be instead explained considering that, even if at higher pH, the decrease in CS protonation degree gave rise to a weakening of its interactions with TPP, and, consequently, to a weakening of the whole NP structure. The net surface charge of CEF molecules remained positive until pH < 7.4 (pKa 2.06 in water at 25 °C) [52] and then they could continue to interact with the polyanionic salt molecules. The slower, more prolonged CEF release profile, poorly affected by changes in gastric pH, as provided by CS NPs, was considered more suitable to obtain a gradual and more reproducible delivery of the drug through the entire gastrointestinal tract.

### 3.5. Stability Studies in Simulated Gastric Fluid

Stability studies in gastric conditions were then performed in order to get more insight and better understand the reasons for the different drug release behavior shown by NLCs and CS NPs. The nanoparticles stability was evaluated in terms of mean size and PDI variations during an incubation period of 2 h at 37 °C in SGF at pH 1.2 or 4.5.

Interestingly, as it appears evident from Figure 2, the different composition of the two investigated nano-systems resulted in a very different behavior when in contact with the gastric environment. CS NPs were stable at both acidic pH values during the whole analyzed time range, showing no significant changes in their size and PDI values. These results confirmed the previous hypothesis that the structure of CS NPs at low pH values is well maintained as a consequence of the strong ionic interactions between CS and TPP.

On the contrary, NLCs showed a very strong increase in size after only 30 min incubation in both gastric media, reaching about 1100 nm after 60 min at pH 1.2 and about 1000 nm after 90 min at pH 4.5. Moreover, the increase in size of the nanoparticles was joined with an increase in their PDI values, indicative of loss of homogeneity of the system. These findings were clearly indicative of a destabilization of the nanodispersions, with the formation of large aggregates reaching the micrometric range, which could be considered responsible for the observed fast and poorly reproducible drug release profile. The dependence of the lipid nanoparticles physical stability on the pH of the medium has already been reported for SLNs (Solid Lipid Nanoparticles) [53]. It has been shown that a decrease of pH coupled with high electrolyte concentrations can result in aggregation phenomena due to a decrease of the electrostatic repulsion among the nanoparticles, attributed to the compression of the diffuse electrical double layer surrounding them [53,54]. Moreover, high electrolyte concentrations combined with low pH values can lead to dehydration of the adsorbed layer of the non-ionic surfactant (such as Pluronic^®^ F68), decreasing its thickness and density and then its steric stabilizing effect, thus further contributing to the destabilization of the nanoparticles [54].

### 3.6. Storage Stability Studies

The selected NLC4_DL4_ and CS NP_DL4_ formulations were stored at 4 ± 1 °C for 3 months and weekly checked for mean particle size, PDI, and Zeta potential, as well as subjected to a visual control to evaluate the possible formation of mold or the appearance of aggregation and/or precipitation phenomena.

Both NLCs and CS NPs essentially retained their mean size and PDI values during the whole period examined, indicating the physical stability of the colloidal dispersions (Figure 3). A different behavior was instead revealed by the surface charge of the nanoparticles. In fact, while in the case of CS NPs, the Zeta potential remained almost unchanged, a progressive clear decrease of its value was found in the case of NLCs. Since the positive charge of NLCs was principally conferred by the cationic lipid stearylamine, the observed reduction in Zeta potential values seems to be attributed to the partial leakage of this component from the NLCs shell during the storage. However, despite the low Zeta potential value reached by NLCs after 3 months storage (about +13 mV), the colloidal dispersion maintained its physical stability, as proved by the almost unchanged size and PDI values, that allowed for the exclusion of the formation of nanometric aggregates that would be invisible to the visual inspection. Evidently, when NLCs were stored as such in their original aqueous medium, the electrostatic repulsion played only a secondary role on their physical stability, and the steric stabilizing effect exerted by the surfactant Pluronic^®^ F68 layer was enough to protect the particles against aggregation phenomena, despite their reduced Zeta potential. Moreover, the visual inspection of the samples did not highlight the presence of mold, and allowed to exclude aggregation or drug crystallization phenomena in any of the samples, thus further confirming their physical stability.

At the end of the storage period, a control of the EE% of the selected NLC4_DL4_ and CS NP_DL4_ formulations was also carried out to determine the entity of drug leakage phenomena. In both cases, only a slight, not significant (*p* > 0.05), reduction was observed that did not exceed 5%, proving the very low tendency of both kinds of nanocarriers to premature drug expulsion. Moreover, this result was also considered indicative of the good chemical stability of the drug in both the selected formulations.

### 3.7. In Vitro Cytotoxicity Assays

Even though all the developed formulations have been prepared starting from biocompatible materials, it is important to gain appropriate information about their actual absence of cytotoxicity. Caco-2 cell line is a largely used in vitro model for cell viability and cytotoxicity studies of oral formulations, and it is widely accepted as a first indicator. The cytotoxicity evaluation of the selected NLC_DL4_ and CS NP_DL4_ formulations and of their corresponding empty formulations and raw materials, including the free drug (all at the same concentrations as in the final formulations) was carried out by two distinct tests: the propidium iodide assay, to evaluate the cell membrane integrity, and the MTT assay, to evaluate the cell metabolic activity.

Figure 4 shows the results obtained with the test of propidium iodide, which is a fluorescent molecule used as dead cells indicator because it cannot penetrate into live cells with intact membranes [55]. The propidium iodide uptake is expressed as relative fluorescence intensity (RFI) with respect to cells incubated with culture medium. As expected, high uptake values of propidium iodide were found in the case of the positive control (SDS), where the damaged cellular membranes become permeable to this dye (RFI > 2); on the contrary, a negligible uptake value (RFI ≈ 1) was observed for the negative control (fresh culture medium), where the membranes of the viable cells remained intact. As for NLCs related samples, both empty and loaded NLCs formulations showed propidium iodide uptake values significantly higher (*p* < 0.01) than the negative control, even though they were always significantly lower (*p* < 0.01) than the positive control. However, after 1:1 dilution, all the samples gave uptake data comparable to the culture medium (*p* > 0.01), resulting in the complete lack of cytotoxicity. On the contrary, all CS NP-related samples exhibited low uptake values, like those of the negative control (*p* > 0.01), without needing any dilution, thus indicating that they did not affect in any way the integrity of the cellular membranes. A possible reason for the results obtained with NLCs could be the presence of the cationic lipid stearylamine on the surface of the lipid nanoparticles. In fact, it has been reported that the incorporation of some cationic additives, such as cetyltrimethylammonium bromide, induced cytotoxicity, and cell death, were attributed to the interaction of the cationic surfactant with the cell membrane, triggering its disruption [56]. However, stearylamine has been successfully incorporated into different kinds of nanocarriers, including NLCs, liposomes, polymeric, or solid lipid nanoparticles, to improve their physicochemical stability, without giving rise to toxic effects [57,58,59].

To support the findings obtained with the propidium iodide test, cytotoxicity studies were also performed by the MTT test (Figure 5). The MTT reagent is reduced by a mitochondrial enzyme and transformed into formazan: the color change is an indicator of the occurrence of the cell metabolic activity and therefore of the cell viability. The reduction of metabolic activity is therefore used as an indicator of cellular damage [60].

As can be seen in Figure 5, a cell viability around 100% was always found for the negative control (fresh culture medium), while for the positive control (SDS, 1 mg/mL), it was only around 12%. An about 100% cell viability was found for all CS NP samples as such, similarly to that found for the negative control (*p* > 0.01), further corroborating their safety in use and full absence of toxicity. On the contrary, a significant reduction (*p* < 0.01) of cell viability to about 60% was observed in the case of the NLC4_DL4_ formulation, and a 1:1 dilution was needed to achieve an about 91% cell viability, which can be considered indicative of absence of cytotoxic events.

## 4. Conclusions

The present work was aimed at developing safe and effective oral liquid formulations of poorly soluble drugs suitable for pediatric use. With this purpose, two different formulations, based on the use of NLCs or CS NPs as drug nanocarriers, have been developed and compared, using CEF as a slightly soluble model drug.

The selected lipid and polymeric nanoparticles formulations showed a similar mean size (around 390 nm) and Zeta potential values much higher than 30 mV, considered as the threshold value assuring good stability against the tendency of nanoparticles towards coalescence [61]. However, NLCs showed lower PDI values than CS NPs (0.30 vs. 0.48), indicative of a better homogeneity of their dispersion. As for EE%, it was only slightly higher for NLCs (36.5 vs. 31.5%), while LE% was clearly higher for CS NPs (5.2 vs. 1.4%).

CS NPs maintained their size, PDI, and Zeta potential values almost without change during 3 months of storage at 4 ± 1 °C; in contrast, NLCs exhibited a marked decrease in Zeta potential, which fell well below the stability threshold value [61]. The most marked differences were observed in release studies, revealing that CS NPs provided a more reproducible and sustained profile than NLCs. These findings related to their different behavior in simulated gastric conditions, where CS NPs were stable at both acidic pH values (1.1 and 4.5), while NLCs suffered a rapid size increase, up to micrometric dimensions, and a concomitant PDI increase, from 0.3 up to 0.6–0.7.

Finally, cytotoxicity studies evidenced the complete biocompatibility of CS NPs, while NLCs required a 1:1 dilution to obtain analogous cell viability values.

In conclusion, based on the overall results, CS NPs were selected as the best CEF nanocarrier, considering their sustained and more reproducible drug release behavior through the gastrointestinal tract, barely influenced by changes in gastric pH, their better stability under storage, and especially, considering that the developed formulations are aimed for pediatric use, with a complete absence of toxicity. However, in regards to this last point, it must be highlighted that the cytotoxicity data obtained for the NLCs should not be regarded as sufficient to discard this formulation, since the in vivo conditions necessarily involve a series of dilution processes as a consequence of the presence of the saliva and the gastric fluids, thus reducing their possible adverse effects.

## Figures and Tables

**Figure 1 pharmaceutics-15-01305-f001:**
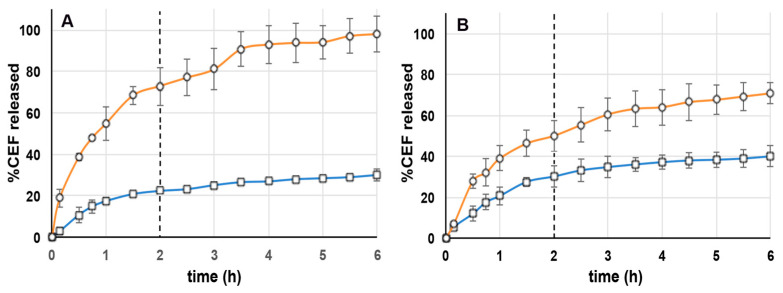
Release profiles of CEF from NLC4_DL4_ (○) and CS NP_DL4_ (□) formulations: (**A**) 2 h in SGF pH 1.2 and 4 h in SIF pH 6.8; (**B**) 2 h in SGF pH 4.5 and 4 h in SIF pH 6.8.

**Figure 2 pharmaceutics-15-01305-f002:**
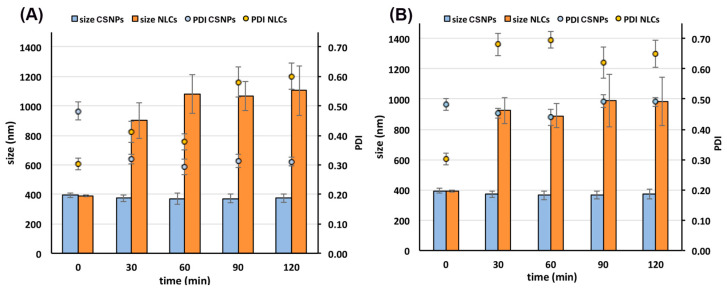
Stability of the selected NLC4_DL4_ and CS NP_DL4_ formulations in SGF at pH 1.2 (**A**) or at pH 4.5 (**B**).

**Figure 3 pharmaceutics-15-01305-f003:**
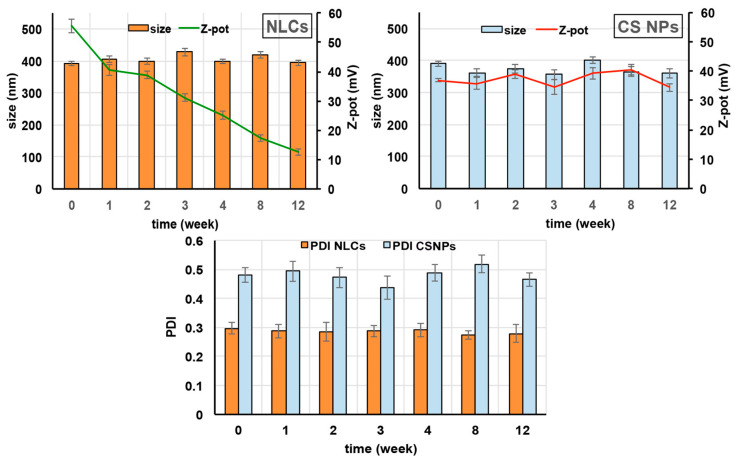
Effect of storage on mean particle size, PDI, and Zeta potential of the selected NLC4_DL4_ and CS NP_DL4_ formulations.

**Figure 4 pharmaceutics-15-01305-f004:**
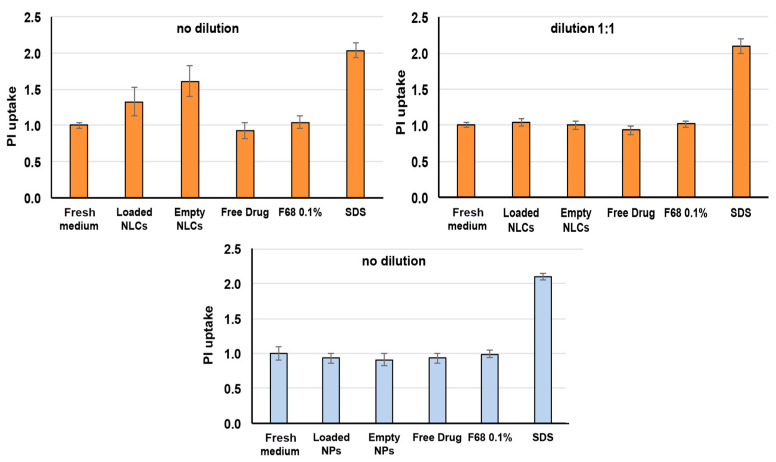
Propidium iodide (PI) uptake by Caco-2 cell line after 24 h incubation with empty and CEF-loaded NLCs (orange bars) or CS NPs (blue bars) and their components. Fresh culture medium and SDS (1 mg/mL) were used as negative and positive control. Results are expressed as relative fluorescence intensity (RFI) with respect to cells incubated with culture medium (mean ± SD, 4 replicates per plate in 3 independent plates).

**Figure 5 pharmaceutics-15-01305-f005:**
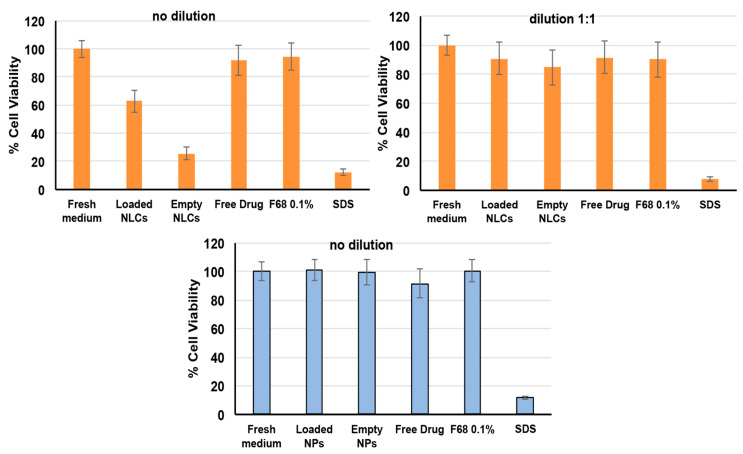
The percent cell viability of Caco-2 cell line measured by MTT test after 24 h incubation with empty and CEF-loaded NLCs (orange bars) or CS NPs (blue bars) and their components. Fresh culture medium and SDS (1 mg/mL) were used as negative and positive control, respectively (mean ± SD, 4 replicates per plate in 3 independent plates).

**Table 1 pharmaceutics-15-01305-t001:** Cefixime solubility (in terms of solution transparency or turbidity) in melted solid lipids (80 °C) and in liquid lipids heated at 80 °C (10 mg drug added to 200 or 500 mg solid lipid or 200 or 500 µL liquid lipid).

Solid Lipid	Solution Appearance 200 mg Lipid 500 mg Lipid		Liquid Lipid	Solution Appearance 200 µL Lipid 500 µL Lipid	
Precirol^®^ ATO5	turbid	turbid	Labrafac^®^ Lipoph	turbid	turbid
Compritol^®^ 888ATO	turbid	turbid	Labrasol^®^	turbid	turbid
Geleol^®^	turbid	turbid	Labrasol^®^ ALF	turbid	turbid
Cetyl palmitate	turbid	turbid	Transcutol^®^ HP	transparent	transparent
Glyceryl tripalmit.	turbid	turbid	Transcutol^®^ P	transparent	transparent
Imwitor^®^ 491	turbid	turbid	Miglyol^®^ 810N	turbid	turbid
Imwitor^®^ 988	turbid	turbid	Miglyol^®^ 812	turbid	turbid
Gelucire^®^ 48/16	turbid	turbid			
Stearylamine	turbid	turbid			

**Table 2 pharmaceutics-15-01305-t002:** Composition of the different NLCs formulations empty (E) or drug-loaded (DL).

Formulation Code	Precirol^®^ ATO5 mg/mL	Gelucire^®^ 48/16 mg/mL	Stearylamine mg/mL	Transcutol^®^ HP µL/mL	Cefixime mg/mL
NLC1_E_	5.0	/	/	5.0	/
NLC1_DL_	5.0	/	/	5.0	0.1
NLC2_E_	10.0	/	/	5.0	/
NLC2_DL_	10.0	/	/	5.0	0.1
NLC3_E_	10.0	20.0	/	5.0	/
NLC3_DL_	10.0	20.0	/	5.0	0.1
NLC4_E_	5.0	/	0.5	5.0	/
NLC4_DL1_	5.0	/	0.5	5.0	0.1
NLC4_DL2_	5.0	/	0.5	5.0	0.2
NLC4_DL4_	5.0	/	0.5	5.0	0.4

**Table 3 pharmaceutics-15-01305-t003:** Physicochemical properties of NLCs in terms of mean size, polydispersity index (PDI), and Zeta potential (Z-pot). Encapsulation efficiency (EE%) and loading efficiency (LE%) of the selected formulations are also reported (see Table 1 for NLC composition).

Formulation Code	Size ± S.D. (nm)	PDI ± S.D.	Z-Pot ± S.D. (mV)	EE% ± S.D.	LE% ± S.D.
NLC1_E_	154.2 ± 2.7	0.23 ± 0.01	−30.7 ± 0.7	/	/
NLC1_DL_	aggregation	/	−9.2 ± 0.4	/	/
NLC2_E_	203.6 ± 5.2	0.18 ± 0.01	−24.8 ± 0.9	/	/
NLC2_DL_	aggregation	/	−9.7 ± 1.3	/	/
NLC3_E_	85.8 ± 8.3	0.57 ± 0.03	−5.5 ± 0.5	/	/
NLC3_DL_	112.4 ± 18.9	0.40 ± 0.07	−5.4 ± 1.1	/	/
NLC4_E_	381.6 ± 5.9	0.34 ± 0.02	+59.9 ± 1.6	/	/
NLC4_DL1_	396.3 ± 3.2	0.28 ± 0.03	+54.4 ± 1.0	39.82 ± 4.01	0.38 ± 0.02
NLC4_DL2_	389.4 ± 4.5	0.32 ± 0.02	+55.5 ± 1.7	36.38 ± 3.09	0.65 ± 0.05
NLC4_DL4_	391.3 ± 7.1	0.30 ± 0.02	+55.6 ± 1.3	36.50 ± 2.33	1.40 ± 0.16

**Table 4 pharmaceutics-15-01305-t004:** Composition of the different empty (E) or drug-loaded (DL) CS NP formulations.

Formulation Code	CS mg/mL	TPP mg/mL	Cefixime mg/mL
NP_E_	2.0	2.5	0.0
NP_DL1_	2.0	2.5	0.1
NP_DL2_	2.0	2.5	0.2
NP_DL4_	2.0	2.5	0.4

**Table 5 pharmaceutics-15-01305-t005:** Properties of CS NPs in terms of mean size, polydispersity index (PDI), Zeta potential, encapsulation efficiency (EE%), and loading efficiency (LE%) (see Table 4 for CS NPs composition).

Formulation Code	Size ± S.D. (nm)	PDI ± S.D.	Z-Pot ± S.D. (mV)	EE% ± S.D.	LE% ± S.D.
NP_E_	412.1 ± 15.3	0.55 ± 0.17	+44.6 ± 1.8	/	/
NP_DL1_	426.5 ± 17.4	0.51 ± 0.03	+42.7 ± 1.0	16.39 ± 1.47	0.7 ± 0.11
NP_DL2_	435.4 ± 30.6	0.52 ± 0.16	+42.3 ± 0.7	21.30 ± 2.00	1.8 ± 0.20
NP_DL4_	395.5 ±16.7	0.48 ± 0.02	+41.9 ± 0.6	31.15 ± 1.57	5.2 ± 0.12

## Data Availability

Data sharing is not applicable to this article.

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
