# Peer review of "Design, Evaluation and Comparison of Nanostructured Lipid Carriers and Chitosan Nanoparticles as Carriers of Poorly Soluble Drugs to Develop Oral Liquid Formulations Suitable for Pediatric Use"

_pharmaceutics, 2023, doi:10.3390/pharmaceutics15041305_

Round 1
Reviewer 1 Report
The paper presents a well performed comparison between two nanosystems with good potential interest for the reader.
My only suggestion is that a few lines about the relevance of drug parameters (LogP and solubility) could be also useful to understand the results of encapsulation and release
Author Response
We thank very much the Reviewer for his positive comments.
As suggested we added few lines in the Introduction about the problems of the very low aqueous solubility of the drug (that is considered the key parameter for its low oral bioavailability), and the possibility of overcoming this issue by the drug entrapment into suitable lipid or polymeric nanoparticles (see lines 90-94).
Reviewer 2 Report
This paper compares two different cefixime formulations which are intended for paediatric patients. The chitosan nanoparticles (CSNPs) and nanostructured lipid carriers (NLCs) were characterised for size, zeta potential and polydispersity. Also tested was release in simulated gastric and intestinal fluid; stability at 4oC. Cytotoxicity in the intestinal cell line, Caco-2 was also performed. The need for oral formulations for paediatrics is an interesting area of study and the paper does importantly carry out the release studies with the pH of the paediatric fluid in mind.
It would be good if the authors outlined what the target profile of these formulations would be. What makes one particle better than the other. Overall the paper characterises the particles well but the pdi is high and the cytotoxicity of the formulations seems to have only been carried out at one concentration.
20-21: reword this sentence. Better to give the actual values for each.
102 “permeation enhancer power” – reword. E,g, permeation enhancing ability.
Why did you test both Labrasol and Labrasol ALF as Labrasol is topical and Labrasol ALF is oral.
130-132 – reword. Was determined as per …
All the formulations were stored at 4°C for further investigations – for how long ? E.g. for no more than a week etc.
Why was UV and not HPLC used to quantify drug ?
Was the weight in mg or ug when calculating EE% and LE% ?
Is ref 27 the correct reference as it is for the hot high-shear homogenization whereas the reference seems different to the method used in this paper and is a solid lipid nanoparticle.
Is the wording correct for the equation for loading efficiency : weight of the empty NLCs or CS NPs. Do you mean the total weight of the nanoparticle or ingredients in the nanoparticle?
232 explain why Pluronic was used?
Section 2.11
Describe culture conditions of the Caco-2 cells and give the passage numbers used.
Why was 24h chosen as the exposure time? Would the intestine be exposed for that long ?
Was MTT and propidium iodide assays carried out in the same experiment on the same cells?
Were these experiments carried out on three separate plates or replicates on the one plate ? n numbers have not been given. Not sure what “various samples to be tested, each of which was analyzed in twelve wells per plate” means. Need to clrify.
323-325 – could you explain this more as Labrasol ALF has the same HLB of 12 and is a non-ionic surfactant. In the table could you add the HLB of each of the lipids tested.
In terms of the NLCs was there a target size, zeta potential and pdi you were looking for ?
364: According to the Gattefosse website the HLB of Gelucire 48/16 is 12 not 16. https://www.gattefosse.com/pharmaceuticals-products/gelucire-4816 and Jannin V, Chevrier S, Michenaud M, Dumont C, Belotti S, Chavant Y, Demarne F. Development of self emulsifying lipid formulations of BCS class II drugs with low to medium lipophilicity. Int J Pharm. 2015 Nov 10;495(1):385-392. doi: 10.1016/j.ijpharm.2015.09.009. Epub 2015 Sep 10. PMID: 26364710.
For the NLCs did you try and reduce the pdi?
Was there a target size, zeta potential and pdi for the chitosan particles ? The pdi seems very high.
Was the size of the chitosan particles measured in the Pluronic F68 0.1% w/v?
Was any imaging performed on the particles ?
Figure 1: might be better to have a and b rather than saying left and right . Same goes for other figures.
What release profile did you want for the NLCs and chitosan NPs. Burst or sustained? In the stomach or intestine?
Did you think about of using Fasted state simulated intestinal fluid ?
Did you test stability in cell culture media as you could have aggreagation of the particles?
Figure 4: n=12 does it mean 12 wells on a plate or n=3 with 4 replicates per plate. A plate is n=1. It is really important that you clarify this.
For cytotoxicity what was the concentration of particles and the drug ? Was the comparison of loaded particles made on the basis of how much of each formulation you would need to achieve x concentration of the drug? Why was this concentration tested?
Did you perform statistics on the cell work ? Comparing to media alone.
593: is the viability for media alone always 100% as everything is relative to that ?
Normally cytotoxicity is measured over a range of concentrations the effects are concentration dependent. Also, by diluting the NLCs you are reducing the concentration ?
Discussion
How do these formulations compare to other cefixime formulations especially for children ?
References:
Many of the references are reviews. More research papers should be included.
No reference for the cell work.
Reviewer 3 Report
The authors describe the formulation of pediatric formulations for a poorly soluble drug (cefixime). The following comments need to be addressed:
1- While the need for pediatric formulations is important, it is not clear why the authors chose chitosan nanoparticles and NLCs for this purpose, especially that their large scale production is difficult.
2- Chitosan nanoparticles are most commonly used for encapsulation of hydrophilic drugs. It is a bit strange that the authors used this system for encapsulation of a class II drug such as cefixime. The authors need to provide references to support the use of chitosan nanoparticles for encapsulation of water insoluble drugs.
3- The authors did not check the stability of cefixime in the two chosen systems upon storage. They only checked the stability of the nanoparticles in terms of physicochemical properties such as particle size, PDI and zeta potential
4- The authors need to unify the abbreviation of nanostructured lipid carriers as either NLC or NLCs throughout the manuscript.
5- The manuscript should be revised by a native speaker
6- The conclusion section is very long. It needs to be shortened to highlight only the significant findings of the manuscript.
Reviewer 4 Report
The present article, (pharmaceutics-2329040), by Giulia Nerli et al, refers to the authors studies on pediatric drug formulations, involving chitosan nanoparticles (CSNPs) and nanostructured lipid carriers (NLCs). The antibiotic drug, Cefixime, was used, as a model bioactive substance, being a abroad scope antibacterial agent used for the treatment of otitis media, strep throat, pneumonia, urinary tract infections, gonorrhea, and Lyme disease. The cytotoxicity experiments confirmed the authors' findings, i.e. that CSNPs as the best nanocarrier, proving their complete biocompatibility, while NLCs formulations needed 1:1 dilution to obtain acceptable cell viability values. It should be pointed out, though, that the cytotoxicity data obtained for the NLCs should not be regarded as sufficient to discard this formulation, since the in vivo conditions necessarily involve a series of dilution processes, as a consequence of the presence of the saliva and the gastric fluids, thus reducing their possible adverse effects. The manuscript is concisely written and the results well documented. The methods used are appropriate and well established. Moreover, this report is of interest to the cognizant reader.
Author Response
We thank very much the Reviewer for his positive comments and we are pleased that he appreciated our work.
Round 2
Reviewer 2 Report
The authors have addressed the reviewers comments.